A novel nomogram to predict the risk of requiring mechanical ventilation in patients with sepsis within 48 hours of admission: a retrospective analysis

Wang Bin 1
Ouyang Jian 1
Xing Rui 2
Jiang Jiyuan 1
Ying Manzhen 1 dm.0579@163.com
1 Emergency Department, Dongyang Hospital Affiliated to Wenzhou Medical University , Jinhua City, Zhejiang , China
2 Haemaology Department, Dongyang Hospital Affiliated to Wenzhou Medical University , Jinhua City, Zhejiang , China
Wang Jincheng
Electronic publication date: 2024 Nov 1
Publication date: 2024
Volume: 12
Electronic Location ID: e18500
Received 2024 Aug 9; Accepted 2024 Oct 18
Copyright: © 2024 Wang et al.
Copyright year: 2024
Copyright holder: Wang et al.
License: This is an open access article distributed under the terms of the Creative Commons Attribution License, which permits unrestricted use, distribution, reproduction and adaptation in any medium and for any purpose provided that it is properly attributed. For attribution, the original author(s), title, publication source (PeerJ) and either DOI or URL of the article must be cited.
License URL: https://creativecommons.org/licenses/by/4.0/

Keywords: Sepsis, Mechanical ventilation, Prediction model, Machine learning

Funding: The authors received no funding for this work.

==============================
Objective

To establish a model that can predict the risk of requiring mechanical ventilation within 48 h after admission in patients with sepsis.

Methods

Data for patients with sepsis admitted to Dongyang People’s Hospital from October 2011 to October 2023 were collected and divided into a modeling group and a validation group. Independent risk factors in the modeling group were analyzed, and a corresponding predictive nomogram was established. The model was evaluated for discriminative power (the area under the curve of the receiver operating characteristic curve, AUC), calibration degree (Hosmer-Lemeshow test), and clinical benefit (decision curve analysis, DCA). Models based on the Sequential Organ Failure Assessment (SOFA) scores, the National Early Warning Score (NEWS) scores and multiple machine learning methods were also established.

Results

The independent factors related to the risk of requiring mechanical ventilation in patients with sepsis within 48 h included lactic acid, pro-brain natriuretic peptide (PRO-BNP), and albumin levels, as well as prothrombin time, the presence of lung infection, and D-dimer levels. The AUC values of nomogram model in the modeling group and validation group were 0.820 and 0.837, respectively. The nomogram model had a good fit and clinical value. The AUC values of the models constructed using SOFA scores and NEWSs were significantly lower than those of the nomogram (P < 0.01). The AUC value of the integrated machine-learning model for the validation group was 0.849, comparable to that of the nomogram model (P = 0.791).

Conclusion

The established nomogram could effectively predict the risk of requiring mechanical ventilation within 48 h of admission by patients with sepsis. Thus, the model can be used for the treatment and management of sepsis.

Introduction

Sepsis is caused by an uncontrolled immune response to infection, resulting in life-threatening organ dysfunction (Fang et al., 2017a, 2017b; Singer et al., 2016), with high morbidity and mortality (Angus et al., 2001; Phua et al., 2011). Due to the increasing incidence in the past few decades (Suarez De La Rica, Gilsanz & Maseda, 2016), sepsis has become one of the main burdens of major diseases worldwide (Fleischmann et al., 2016; Gaieski et al., 2013; Iwashyna et al., 2012; Vincent et al., 2014). After implementing the “Surviving Sepsis Campaign: International guidelines for the management of severe sepsis and septic shock” (Bouferrache et al., 2012; Dellinger et al., 2013), more patients with sepsis have been successfully treated, leading to a decrease in related mortality and disability rates. However, our previous data showed that about 20% of patients with sepsis develop respiratory failure (Wang, Chen & Wang, 2023), which may be associated with uncontrolled inflammation, fluid resuscitation, and abnormal clotting. Respiratory failure significantly increases the mortality rate of patients with sepsis and, thus, has become an independent risk factor in multiple prognostic predictive models (Angus et al., 2001; Bellani et al., 2016; Cochi et al., 2016; Fowler et al., 2019).

Early mechanical ventilation is one of the pivotal treatments for patients with respiratory failure. Studies have shown that early mechanical ventilation improves the prognosis of patients with sepsis (Fang et al., 2020). Currently, the Sequential Organ Failure Assessment (SOFA) (Vincent et al., 1996) and the National Early Warning Score (NEWS) (Subbe et al., 2001) are the main models used clinically to predict the severity of sepsis, and efficiently predict the risk of organ failure in this group of patients. Six indicators comprise the SOFA, with higher scores indicating a worse prognosis for patients with sepsis (Vincent et al., 1996). The patient’s respiratory rate, consciousness, heart rate, systolic pressure, and temperature are included in the NEWS (Subbe et al., 2001). Because of their early establishment, an increasing number of sepsis-related indicators were not included. Moreover, these scores were not specifically developed to predict the risk of requiring mechanical ventilation in patients with sepsis.

Multiple current models can predict the risk of acute respiratory distress (ARDS) in patients with sepsis (Bai et al., 2022; Xu et al., 2023). With more variables being related to ARDS, machine-learning models have been adopted to predict patients’ prognosis (Bai et al., 2022; Jiang et al., 2024). Researchers have proposed several methods for machine learning, such as the Decision Tree Algorithm (C5.0), Xtreme Gradient Boosting (XGBOOT), and the support vector machine (SVM). Recently, an ensemble method developed by combining several methods has been shown to overcome the disadvantages caused by a single method and to efficiently predict outcomes with a higher discriminatory power (Bannick, McGaughey & Flaxman, 2021; Bhagat, Tung & Yaseen, 2021; Karalar, Kapucu & Guruler, 2021; Peppes et al., 2021). The many predictive factors in the machine-learning model might cause overfitting. However, not all ARDS patients need mechanical ventilation, and no model can predict the short-term need for mechanical ventilation in patients with sepsis.

Thus, herein, we established a model that can predict whether patients with sepsis will need mechanical ventilation within 48 h after admission. This model is useful in clinical decision-making by healthcare providers.

Materials and Methods

Patient definition, inclusion, and exclusion criteria

Data for 2,044 patients with sepsis admitted to the Dongyang People’s Hospital between October 2011 and October 2023 were collected and analyzed. For inclusion in the study, the patients had to meet the diagnostic criteria of sepsis 3.0: with an infection that has caused the SOFA score to increase by two points. The exclusion criteria were: 1. patients younger than 18 years old, 2. patients with respiratory failure upon admission (patients with oxygenation index less than 300 mmHg and pulse oxygen saturation (SPO2) less than 90% upon admission), 3. patients who did not finish treatment and those who met the indications for mechanical ventilation but refused to have this intervention, and 4. patients who underwent emergency surgery.

This study was approved by the Ethics Committee of Dongyang People’s Hospital (approval number 2024-YX-096). The data was anonymously analyzed after removing personal identification information. The study was carried out in accordance with the principles of the Declaration of Helsinki and its amendments.

Retrospective data collection from medical records

Patients’ information, including gender, age, and medical history, including hypertension, type 2 diabetes mellitus, cerebral infarction, chronic lung disease (including chronic obstructive pulmonary disease (COPD)), interstitial pneumonia, and chronic pulmonary fibrosis, chronic heart disease (including cardiac dysfunction with New York Heart Association (NYHA) II grade and above), chronic liver disease (cirrhosis), chronic kidney disease (chronic renal insufficiency and nephrotic syndrome), leukemia, and malignancy status, and other serum parameters status, such as high-sensitivity C-reactive protein (HS-CRP), alanine transaminase, triglyceride, total bilirubin, creatinine, lactic acid (LA), pro-brain natriuretic peptide (PRO-BNP), cholinesterase, prothrombin time (PT), D-dimer, potassium, sodium, magnesium, calcium, hemoglobin (HB), albumin, and globulin levels, as well as white blood cell (WBC) and platelet counts, were extracted from patient records. Vital signs upon admission, including SPO2, temperature, mean arterial pressure (MAP), heart rate (HR), respiratory rate, the Glasgow coma score (GCS), and infection sites, such as intracranial, lung, digestive tract, biliary tract, and urinary tract, were also recorded. The indications for the need for mechanical ventilation were as follows: 1. after active interventions such as drug oxygen inhalation, high-flow oxygen therapy, and non-invasive ventilator, the patient still exhibited obvious dyspnea, chest tightness, and shortness of breath, with a respiratory rate of >35 times/min, oxygen partial pressure of less than 50 mmHg or oxygenation index of <200; 2. respiratory depression occurred, and the number of breaths was less than 8/min; 3. patients who developed severe consciousness disorders, including lethargy and coma; and 4. a progressive increase in the partial pressure of carbon dioxide, accompanied by a pH of 7.20 or lower (mechanical ventilation was evaluated by physicians).

Variable screening and predictive model construction

Partially continuous variables were converted to categorical variables based on clinical normal ranges to facilitate clinical interpretation. The data were randomly divided into the modeling group and the validation group at a ratio of 7:3. Following univariate analysis of the modeling group data, multiple collinear analysis (indicated by the variance inflation factor, VIF) was conducted to confirm that no collinearities existed between any two included variables (Hsieh & Lavori, 2000). Then, a linear relationship between the variables and logitP was confirmed using the BoxTidwell function (P ≥ 0.05). The criteria for the above analysis were provided in our previous work (Wang, Chen & Wang, 2023). Finally, a multi-variable logistic regression analysis was performed, followed by a stepwise regression analysis to identify independent risk factors and establish a predictive nomogram.

Accuracy and reliability of the nomogram

A receiver operating characteristic (ROC) curve, calibration graph, and decline curve analysis (DCA) were used to evaluate the discriminatory power, goodness of fit, and clinical accuracy of the model, respectively. An area under the curve (AUC) of >0.75 indicated good discrimination ability (Silva et al., 2015). The cut-off value was determined based on the maximal Youden indexes in the ROC. The sensitivity, specificity, prediction accuracy (ACC), negative predictive value (NPV), and positive predictive value (PPV) of the model were also determined. A calibration graph was used to evaluate the consistency between predicted and observed probabilities. A P-value greater than 0.05 in the Hosmer-Lemeshow test indicated an acceptable goodness of fit of the model (Niu et al., 2017). Finally, the model was considered to exert good clinical validity when the model’s curve was far from the two extreme curves (Nattino, Finazzi & Bertolini, 2016).

Comparing the accuracy of the nomogram and other models

Two models were constructed based on SOFA scores and NEWSs to compare the predictive power of the established nomogram. Several machine-learning models were also constructed in the modeling group and validated in the validation group. The methods for the machine-learning models included C5.0, XGBOOT, and SVM. An integrated model was also established using the stacking method (Jiang et al., 2022b). The DeLong test was performed to compare the discriminatory power of our nomogram with those of other models.

Statistical analysis

Normally distributed continual variables were expressed as X¯ ± s, and differences between corresponding groups were analyzed using the two independent-samples t-test method. Non-normally distributed continual data were expressed as the median (quartiles), and differences between corresponding groups were analyzed using the Mann-Whitney U test. Count data were expressed as rates and percentages, and differences were compared using the χ2 test. All statistical analyses were performed using R (software version 4.1.2) containing packages as described previously (Wang, Chen & Wang, 2023). Statistical significance was set at P < 0.05.

Results

Baseline data of the enrolled participants

A total of 2,336 patients with sepsis admitted to the Dongyang People’s Hospital between 2011 and 2023 were recruited in this study. After excluding 150 patients with respiratory failure at admission, one patient under 18 years old, 131 patients with acute abdominal surgery, and 10 patients who discontinued treatment, 2,044 patients were included in the final analysis. Of the 2,044, 1,431 patients were enrolled in the modeling group, and 613 patients were included in the validation group (Fig. 1). Among them, 108 patients were intubated and placed on a ventilator within 48 h after admission (108/2,044; 5.3%). Of the included patients, 25% had lung infections, 10% patients had bile duct infections, 17% had urinary duct infections, 17% had gastrointestinal infections, and 31% had infections in other anatomic sites or unknown sites. No significant difference was observed in any variable between the modeling group and the validation group (P > 0.05, Table 1).

Figure 1 The inclusion and exclusion process of enrolled participants in this study.

Table 1 Baseline characteristics of the modeling population and validation populationa.

Variables	Total (n = 2,044)	Training (n = 1,431)	Testing (n = 613)	p	
Gender				0.923	
Male	1,052 (51)	735 (51)	317 (52)		
Female	992 (49)	696 (49)	296 (48)		
Age (years)	71 (58, 80)	71 (58, 80)	71 (58, 80)	0.921	
HS-CRP (mg/L)	109.72 (45.56, 178.6)	109.32 (48.01, 178.5)	109.92 (40.4, 180.4)	0.424	
Alanine transaminase (U/L)	24 (14, 43)	24 (14, 43)	24 (14, 44)	0.818	
Triglyceride (mmol/L)	3.09 (2.57, 3.72)	3.07 (2.55, 3.7)	3.11 (2.6, 3.76)	0.174	
Total bilirubin (umol/L)	11.9 (8, 20.7)	11.9 (8.1, 20.2)	12 (7.9, 21.6)	0.888	
Creatinine (umol/L)	96 (71, 145)	96 (70, 145)	97 (71, 144)	0.982	
Lactic acid (mmol/L)	1.9 (1.2, 2.9)	1.9 (1.3, 3)	1.9 (1.2, 2.9)	0.867	
PRO-BNP (pg/ml)	1,201 (441.95, 3,268.75)	1,200 (437.35, 3,251)	1,207 (469.5, 3,292)	0.809	
Cholinesterase (U/L)	4,321 (3,210, 5,460.25)	4,265 (3,184.5, 5,440.5)	4,397 (3,258, 5,493)	0.279	
Prothrombin time (s)	14.9 (14, 16.1)	14.9 (14, 16.2)	14.8 (13.9, 16)	0.072	
D-dime (mg/L)	2.68 (1.46, 5.34)	2.7 (1.48, 5.2)	2.65 (1.43, 5.43)	0.566	
Potassium (mmol/L)				0.593	
3.5–5.5	1,246 (61)	862 (60)	384 (63)		
<3.5	753 (37)	537 (38)	216 (35)		
>5.5	45 (2)	32 (2)	13 (2)		
Sodium (mmol/L)				0.512	
135–145	1,039 (51)	739 (52)	300 (49)		
<135	944 (46)	651 (45)	293 (48)		
>145	61 (3)	41 (3)	20 (3)		
Magnesium (mmol/L)				0.154	
0.75–1.25	1,366 (67)	948 (66)	418 (68)		
<0.75	669 (33)	479 (33)	190 (31)		
>1.25	9 (0)	4 (0)	5 (1)		
Calcium (mmol/L)				1.000	
2.25–2.75	103 (5)	72 (5)	31 (5)		
<2.25	1,938 (95)	1357 (95)	581 (95)		
>2.25	3 (0)	2 (0)	1 (0)		
White blood cell (*10−9/L)				0.599	
4–10	664 (32)	469 (33)	195 (32)		
<4	186 (9)	135 (9)	51 (8)		
>10	1,194 (58)	827 (58)	367 (60)		
Hemoglobin (*10−9/L)				0.331	
110-160	1,174 (57)	808 (56)	366 (60)		
<110	817 (40)	587 (41)	230 (38)		
>160	53 (3)	36 (3)	17 (3)		
Platelet (*10−9/L)				0.780	
100–300	1,431 (70)	1,007 (70)	424 (69)		
<100	483 (24)	332 (23)	151 (25)		
>300	130 (6)	92 (6)	38 (6)		
Albumin (g/L)	29.8 (26.6, 32.7)	29.8 (26.6, 32.75)	30.1 (26.7, 32.7)	0.603	
Globulin (g/L)	26.2 (23.2, 29.6)	26.3 (23.2, 29.8)	26.1 (23.1, 29.2)	0.236	
SPO2 (%)	97 (95, 98)	97 (95, 98)	97 (95, 98)	0.099	
Temperature (°C)				0.578	
36–37.5	845 (41)	589 (41)	256 (42)		
<36	93 (5)	61 (4)	32 (5)		
>37.5	1,106 (54)	781 (55)	325 (53)		
MAP (mmHg)				0.990	
70–105	1,382 (68)	968 (68)	414 (68)		
<70	425 (21)	298 (21)	127 (21)		
>105	237 (12)	165 (12)	72 (12)		
Heart rate (times/min)	98 (85, 114)	98 (86, 114)	98 (84, 112)	0.342	
Breath rate (times/min)	20 (18, 20)	20 (18, 20)	20 (20, 20)	0.269	
GCS	15 (15, 15)	15 (15, 15)	15 (15, 15)	0.435	
Diabetes				0.134	
No	1,641 (80)	1,136 (79)	505 (82)		
Yes	403 (20)	295 (21)	108 (18)		
Hypertension				0.874	
No	1,190 (58)	831 (58)	359 (59)		
Yes	854 (42)	600 (42)	254 (41)		
Cerebral infarction				1.000	
No	1,969 (96)	1,378 (96)	591 (96)		
Yes	75 (4)	53 (4)	22 (4)		
Cancer				0.180	
No	1,694 (83)	1,175 (82)	519 (85)		
Yes	350 (17)	256 (18)	94 (15)		
Chronic lung disease				0.799	
No	1,969 (96)	1,377 (96)	592 (97)		
Yes	75 (4)	54 (4)	21 (3)		
Chronic heart disease				0.458	
No	2,009 (98)	1,404 (98)	605 (99)		
Yes	35 (2)	27 (2)	8 (1)		
Chronic liver disease				0.227	
No	1,948 (95)	1,358 (95)	590 (96)		
Yes	96 (5)	73 (5)	23 (4)		
Chronic kidney disease				1.000	
No	1,985 (97)	1,390 (97)	595 (97)		
Yes	59 (3)	41 (3)	18 (3)		
Leukemia				0.898	
No	2,018 (99)	1,412 (99)	606 (99)		
Yes	26 (1)	19 (1)	7 (1)		
Intracranial infection				0.675	
No	2,038 (100)	1,426 (100)	612 (100)		
Yes	6 (0)	5 (0)	1 (0)		
Lung infection				0.700	
No	1,534 (75)	1,070 (75)	464 (76)		
Yes	510 (25)	361 (25)	149 (24)		
Biliary infection				0.563	
No	1,854 (91)	1,294 (90)	560 (91)		
Yes	190 (9)	137 (10)	53 (9)		
Urinary infection				0.214	
No	1,672 (82)	1,181 (83)	491 (80)		
Yes	372 (18)	250 (17)	122 (20)		
Gastrointestinal infection				0.214	
No	1,672 (82)	1,181 (83)	491 (80)		
Yes	372 (18)	250 (17)	122 (20)		
News score	3 (2, 4)	3 (2, 5)	3 (2, 4)	0.353	
Sofa score	5 (3, 6)	5 (3, 6)	5 (3, 6)	0.805	
Notes:

Callout: a, Continuous variables are described as means and quarterbacks due to non-normal distribution. Categories varies are analyzed by χ2 test and continuous variables are analyzed by Wilcoxon rank sum test; b, first examination index following admission.

HS-CRP, high sensitivity-C reactive protein; pro-BNP, pro-brain natriuretic peptide; SPO2, pules oxygen saturation; MAP, mean arterial pressure; GCS, Glasgow coma score.

Independent factors that predict the need for mechanical ventilation in patients with sepsis

Univariate analysis showed that 11 variables, including triglyceride, creatinine, LA, PRO-BNP, and cholinesterase levels, PT, D-dimer level, platelet count, albumin level, breathing rate, and lung infection (Table 2), were associated with the need for mechanical ventilation in patients with sepsis after admission (P < 0.001). The VIF values for each variable were less than 5, indicating no multicollinearity (Table S1). A linear relationship existed between each continuous variable and logitp (Table S2). The multi-variable logistic regression and stepwise regression analyses showed that six variables, including LA, PRO-BNP, albumin, and D-dimer levels, as well as PT and lung infection, were independent risk factors that predicted the probability of the need for mechanical ventilation in patients with sepsis within 48 h after admission (Table 3).

Table 2 Univariate analysis between ventilation and no ventilation in training populationa.

Variables	Total (n = 1,431)	No ventilation (n = 1,361)	Ventilation (n = 70)	p	
Gender				0.893	
Male	735 (51)	698 (51)	37 (53)		
Female	696 (49)	663 (49)	33 (47)		
Age (years)	71 (58, 80)	71 (58, 81)	65 (52.25, 76.5)	0.017	
HS-CRP (mg/L)	109.32 (48.01, 178.5)	108.7 (47.83, 174.1)	128.16 (51.77, 200)	0.166	
Alanine transaminase
(U/L)	24 (14, 43)	23 (14, 43)	27 (15.25, 59.25)	0.145	
Triglyceride (mmol/L)	3.07 (2.55, 3.7)	3.09 (2.57, 3.72)	2.7 (2.01, 3.15)	<0.001	
Total bilirubin (umol/L)	11.9 (8.1, 20.2)	11.8 (8, 19.8)	15.5 (8.4, 44.1)	0.008	
Creatinine (umol/L)	96 (70, 145)	94 (70, 141)	128.5 (81.5, 237.25)	<0.001	
Lactic acid (mmol/L)	1.9 (1.3, 3)	1.8 (1.2, 2.9)	3 (1.7, 5.2)	<0.001	
Pro-BNP (pg/ml)	1,200 (437.35, 3,251)	1,139 (425.8, 3,032)	3,667.5 (1,457.25, 12,023)	<0.001	
Cholinesterase (U/L)	4,265 (3,184.5, 5,440.5)	4,320 (3,258, 5,465)	3,499 (2,429.75, 4,543.75)	<0.001	
Prothrombin time (s)	14.9 (14, 16.2)	14.8 (14, 16.1)	16.45 (14.77, 19.75)	<0.001	
D-dimer (mg/L)	2.7 (1.48, 5.2)	2.65 (1.44, 5)	5.22 (2.17, 12.53)	<0.001	
Potassium (mmol/L)				0.010	
3.5–5.5	862 (60)	821 (60)	41 (59)		
<3.5	537 (38)	514 (38)	23 (33)		
>5.5	32 (2)	26 (2)	6 (9)		
Sodium (mmol/L)				0.630	
135–145	739 (52)	699 (51)	40 (57)		
<135	651 (45)	623 (46)	28 (40)		
>145	41 (3)	39 (3)	2 (3)		
Magnesium (mmol/L)				0.257	
0.75–1.25	948 (66)	908 (67)	40 (57)		
<0.75	479 (33)	449 (33)	30 (43)		
>1.25	4 (0)	4 (0)	0 (0)		
Calcium (mmol/L)				0.126	
2.25–2.75	72 (5)	72 (5)	0 (0)		
<2.25	1,357 (95)	1,287 (95)	70 (100)		
>2.25	2 (0)	2 (0)	0 (0)		
White blood cell
(*10−9/L)				0.014	
4–10	469 (33)	444 (33)	25 (36)		
<4	135 (9)	122 (9)	13 (19)		
>10	827 (58)	795 (58)	32 (46)		
Hemoglobin (*10−9/L)				1.000	
110–160	808 (56)	768 (56)	40 (57)		
<110	587 (41)	558 (41)	29 (41)		
>160	36 (3)	35 (3)	1 (1)		
Platelet (*10−9/L)				<0.001	
100–300	1,007 (70)	971 (71)	36 (51)		
<100	332 (23)	304 (22)	28 (40)		
>300	92 (6)	86 (6)	6 (9)		
Albumin (g/L)	29.8 (26.6, 32.75)	29.9 (26.8, 32.9)	26.6 (22.83, 30.25)	<0.001	
Globulin (g/L)	26.3 (23.2, 29.8)	26.4 (23.4, 29.8)	24.7 (20.1, 28.35)	0.001	
SPO2 (%)	97 (95, 98)	97 (95, 98)	97 (95, 98)	0.232	
Temperature (°C)				0.395	
36–37.5	589 (41)	562 (41)	27 (39)		
<36	61 (4)	56 (4)	5 (7)		
>37.5	781 (55)	743 (55)	38 (54)		
MAP (mmHg)				0.003	
70–105	968 (68)	931 (68)	37 (53)		
<70	298 (21)	272 (20)	26 (37)		
>105	165 (12)	158 (12)	7 (10)		
Heart rate (times/min)	98 (86, 114)	98 (86, 114)	105.5 (88, 119.5)	0.014	
Breath rate (times/min)	20 (18, 20)	20 (18, 20)	20 (20, 22)	<0.001	
GCS	15 (15, 15)	15 (15, 15)	15 (15, 15)	0.004	
Diabetes				0.016	
No	1,136 (79)	1,072 (79)	64 (91)		
Yes	295 (21)	289 (21)	6 (9)		
Hypertension				0.014	
No	831 (58)	780 (57)	51 (73)		
Yes	600 (42)	581 (43)	19 (27)		
Cerebral infarction				0.108	
No	1,378 (96)	1,308 (96)	70 (100)		
Yes	53 (4)	53 (4)	0 (0)		
Cancer				0.203	
No	1,175 (82)	1,122 (82)	53 (76)		
Yes	256 (18)	239 (18)	17 (24)		
Chronic lung disease				0.515	
No	1,377 (96)	1,308 (96)	69 (99)		
Yes	54 (4)	53 (4)	1 (1)		
Chronic heart disease				0.384	
No	1,404 (98)	1,336 (98)	68 (97)		
Yes	27 (2)	25 (2)	2 (3)		
Chronic liver disease				0.023	
No	1,358 (95)	1,296 (95)	62 (89)		
Yes	73 (5)	65 (5)	8 (11)		
Chronic kidney disease				0.449	
No	1,390 (97)	1,323 (97)	67 (96)		
Yes	41 (3)	38 (3)	3 (4)		
Leukemia				0.617	
No	1,412 (99)	1,343 (99)	69 (99)		
Yes	19 (1)	18 (1)	1 (1)		
Intracranial infection				0.222	
No	1,426 (100)	1,357 (100)	69 (99)		
Yes	5 (0)	4 (0)	1 (1)		
Lung infection				<0.001	
No	1,070 (75)	1,034 (76)	36 (51)		
Yes	361 (25)	327 (24)	34 (49)		
Biliary infection				0.359	
No	1,294 (90)	1,228 (90)	66 (94)		
Yes	137 (10)	133 (10)	4 (6)		
Urinary infection				0.577	
No	1,181 (83)	1,121 (82)	60 (86)		
Yes	250 (17)	240 (18)	10 (14)		
Gastrointestinal infection				0.577	
No	1,181 (83)	1,121 (82)	60 (86)		
Yes	250 (17)	240 (18)	10 (14)		
Notes:

Callout: a, Continuous variables are described as means and quarterbacks due to non-normal distribution. Categories varies are analyzed by χ2 test and continuous variables are analyzed by Wilcoxon rank sum test; b, first examination index following admission.

HS-CRP, high sensitivity-C reactive protein; pro-BNP, pro-brain natriuretic peptide; SPO2, pules oxygen saturation; MAP, mean arterial pressure; GCS, Glasgow coma score.

Table 3 Multivariate logistic regression analysis and stepwise regression analysis of involved variables in modeling group.

Variables	Multivariable logistic regression	Stepwise regression	
OR (95% CI)	P value	OR (95% CI)	P value	
Triglyceride (mmol/L)	0.732 [0.510–1.035]	0.084	0.767 [0.550–1.056]	0.111	
Total bilirubin (U/L)	1.000 [0.997–1.001]	0.433	NA	NA	
Lactic acid (mol/L)	1.103 [1.006–1.197]	0.024	1.108 [1.010–1.201]	0.020	
Pro-BNP (pg/ml)a	1.000 [1.000–1.000]	<0.001	1.000 [1.000–1.000]	<0.001	
Cholinesterase (U/L)	1.000 [0.999–1.000]	0.374	NA	NA	
Prothrombin time (s)	1.026 [1.000–1.056]	0.044	1.027 [1.002–1.057]	0.028	
D-dimer (mg/L)	1.061 [1.016–1.106]	0.006	1.061 [1.017–1.105]	0.005	
Albumin (g/L)	0.925 [0.863–0.990]	0.026	0.934 [0.884–0.987]	0.015	
Breath rate (times/min)	1.023 [0.965–1.068]	0.341	NA	NA	
Platelet <100 (*10−9/L)	1.411 [0.796–2.470]	0.232	NA	NA	
Platelet >300 (*10−9/L)	1.635 [0.549–4.100]	0.330	NA	NA	
Lung infection	2.823 [1.673–4.767]	<0.001	2.749 [1.639–4.611]	<0.001	
Note:

a The exact pro-BNP in the multivariable logistic regression was 1.0000578 (1.00003011–1.0000845); the exact value for pro-BNP in the stepwise regression was 1.0000515 (1.00002786–1.0000739).

Establishment of the nomogram

The nomogram was developed by combining six variables. Each variable was assigned a specific score by vertically mapping the input variables to the top score line. The risk for requiring mechanical ventilation could be obtained by mapping the summed score to the risk line (Fig. 2).

Figure 2 The nomogram model based on logistic regression.

The discriminatiory power, goodness of fit, and clinical accuracy of the model for data in the training set

The AUC value of the model at the 95% confidence interval (CI) was 0.820 [range 0.7689–0.870], demonstrating its good discriminatory power (Fig. 3A). The optimal cut-off score was 0.055, while the specificity score was 0.795 (95% CI [0.772–0.816]), and the sensitivity score was 0.743 (95% CI [0.643–0.843]). The ACC value at the 95% CI was 0.792 [range 0.792–0.793], the PPV value at the 95% CI was 0.157 [range 0.118–0.196], and the NPV value at the 95% CI was 0.984 [range 0.976–0.991]. The P-value of the calibration graph was 0.695, with a Brier value of 0.043 and a slope of 1, indicating a good fit (Fig. 3B). In the DCA curve, the curve of the model was very far from the two extreme curves, indicating that the model had good clinical benefits (Fig. 3C).

Figure 3 Evaluation of prediction model based on the logistic regression in modeling group.

(A) The discrimination power by AUC; (B) the goodness of fit by calibration curve; (C) the clinical benefit by decision curve.

Validation of the model using patients in the validation group

The AUC value for patients in the validation group at the 95% CI was 0.837 [range 0.776–0.898] (Fig. 4A). The optimal cut-off value was 0.043, with a specificity score of 0.642 (95% CI [0.604–0.684]) and a sensitivity score of 0.895 (95% CI [0.790– 0.974]). The ACC value at the 95% CI was 0.657 [range 0.657–0.658], the PPV value at the 95% CI was 0.142 [range 0.098–0.186], and the NPV value at the 95% CI was 0.989 [range 0.979–1]. The P-value for the calibration graph was 0.693, with a Brier value of 0.043 and a slope of 1, indicating a good fit (Fig. 4B). The curve of the model was far from the two extreme curves, demonstrating good discriminatory power, calibration, and clinical accuracy (Fig. 4C).

Figure 4 Validation of prediction model based on the logistic regression in validation group.

(A) The discrimination power by AUC; (B) the goodness of fit by calibration curve; (C) the clinical benefit by decision curve.

Comparison of the nomogram and the models established based on SOFA scores and NEWSs

The AUC value of the predictive model established in the modeling group based on SOFA scores was 0.696 (95% CI [0.631–0.761]), while that established based on the NEWSs was 0.626 (95% CI [0.560–0.690]). Further analysis revealed that the discriminatiory power of our nomogram was higher than that of the model developed based on SOFA scores and NEWSs (the P-values in DeLong’s test were 0.001 and <0.001, respectively (Table 4, Fig. 5A). The AUC value of the model established using SOFA scores in the validation group was 0.664 (95% CI [0.565–0.763]), and the AUC value of the model established using NEWSs was 0.595 (95% CI [0.504–0.687]), significantly lower than that of the logistic model (Fig. 5B).

Table 4 The comparison of AUCs of logistic model and other models in the validation population.

Method	AUC (95% CI)	P	
SVM	0.744 [0.654−0.835]	0.096	
C5.0	0.809 [0.729−0.889]	0.587	
Xgboost	0.867 [0.812−0.923]	0.496	
Ensemble	0.849 [0.751−0.891]	0.873	
Sofa	0.664 [0.565−0.763]	0.002	
News	0.595 [0.504−0.687]	<0.001	
Note:

Callout: SVM, Support Vector Machine; C5.0, C5.0 decision tree algorithm; Xgboost, eXtreme Gradient Boosting.

Figure 5 The discrimination powers of models based on the SOFA and NEWs score.

Comparison with other machine-learning models

The AUCs of machine-learning models in the training model varied with the method, including 0.999 (95% CI [0.998−1.000]) for C5.0, 0.820 (95% CI [0.764−0.876]) for SVM, and 0.994 (95% CI [0.987−1.000]) for XGBoost. The integrated model developed by combining the three machine-learning models was 0.996 (95% CI [0.994−0.999], Fig. S1A). The predicted lines were not close to the ideal line (Fig. S1B), indicating the bad fit of these four methods.

The AUC of the C5.0 model in the validation group was 0.809 (95% CI [0.729–0.889], Fig. 6A), and that of SVM was 0.744 (95% CI [0.654–0.835]). The AUC value of the model developed using XGBoost was 0.867 (95% CI [0.812–0.923]), and the AUC value of the integrated model developed by combining the three machine-learning models was 0.849 (95% CI [0.783–0.908]). The calibration graph indicated that the calibration of each machine-learning model was relatively poor (Fig. 6B). In the validation group, the AUC value of the logistic model was comparable to that developed by Xgboost and the ensemble but higher than that developed using C5.0, SVM, SOFA scores, and NEWSs (Table 4).

Figure 6 The machine learning models based on C5.0, SVM, XGboost and ensemble methods in the validation group.

Discussion

This research study showed that LA, PRO-BNP, D-dimer, and albumin levels, as well as PT and the presence of pulmonary infection, were independent factors that predicted the need for mechanical ventilation in patients with sepsis within 48 h of admission. The above variables were combined into a regression model to develop a predictive nomogram. The model had good discriminatory power, calibration, and clinical accuracy. The discrimination ability of the model was higher than that developed based on SOFA scores and NEWSs and was comparable to that of the integrated model. These variables are common clinical indicators, facilitating easy and convenient application in clinical settings.

LA is an index of aerobic metabolism, and research has shown that LA levels predict sepsis prognosis (Jiang et al., 2022a). Albumin levels predict both short- and long-term sepsis prognosis (Cao et al., 2023). Some studies have shown that low albumin levels at admission increase the risk of respiratory failure, which may be associated with ventilator atrophy and a decrease in plasma osmotic pressure caused by low albumin levels (Blunt, Nicholson & Park, 1998; Thongprayoon et al., 2020; Wang, Chen & Wang, 2023). A higher D-dimer level indicates the occurrence of thrombus in the micro-vessels or large blood vessels of the lung, which can lead to an imbalance in ventilatory blood flow. Recent studies showed that high D-dimer levels are strongly linked to respiratory failure (Takeshita et al., 2021). PRO-BNP, an indicator of cardiac function, can increase due to reasons such as poor cardiac function, renal failure, and excessive fluid replacement, and high PRO-BNP levels can cause respiratory failure (Cheng et al., 2015; Takasu et al., 2013). High PT levels are associated with the development of sepsis and patient prognosis (Kim et al., 2021) and indicate the worsening of sepsis and the occurrence of hypocoagulation, which is closely related to multiple organ dysfunction (Walsh et al., 2010). High PT levels are probably caused by microvascular thrombosis, further disrupting air and oxygen exchange between the blood and tissues. Lung infection itself can damage the lung mucosa and alveoli, cause immune dysfunction, and impair the functioning of macrophages and monocytes, increasing the risk of secondary infection and respiratory failure (Bouras, Asehnoune & Roquilly, 2018; Roquilly et al., 2020). Previous studies considered the above indicators to be risk factors for respiratory failure in patients with sepsis, which demonstrates the reliability of our study results.

Compared to models established based on SOFA scores and NEWSs, our model offers higher discrimination power. This phenomenon could be partially explained by the fact that our predictive model contains variables that are not contained in SOFA scores and NEWSs. Although these variables are commonly tested during sepsis diagnosis and prognosis evaluation, their combination to predict the need for mechanical ventilation has not been reported. Although these indicators may offer some disadvantages, examination is necessary, given the significance of predicting the risk of mechanical ventilation in patients with sepsis.

The discriminatory power of our nomogram in the validation group was compared with machine-learning models, including C5.0, XGBOOT, SVM, and an integrated method. The predictive accuracy for machine learning was sub-optimal, with low fitness between the predicted and observed values. Additionally, more variables were included in the machine-learning models, which substantially decreased its convenience in clinical practice. For example, 14 indicators were included in the integrated model (Fig. S2). Based on the black box theory, findings of the machine-learning model are often difficult to interpret in clinical settings. Thus, machine-learning models are only preferred if they are significantly higher than those developed through traditional modeling (Kelly et al., 2019).

This predictive model was established in patients with sepsis defined by the Sepsis-3 criteria. Other studies used different inclusion criteria defined by Sepsis-2 (Phua et al., 2011; Suarez De La Rica, Gilsanz & Maseda, 2016; Takasu et al., 2013). The definition of Sepsis-2 includes infection and systemic inflammatory response syndrome, while Sepsis-3 includes infection and increases in SOFA scores of two points (Singer et al., 2016). The different definitions used would result in heterogeneity in the included cases, which might influence the predictive power of the established models. Thus, the nomogram model in this study would be suitable for patients with sepsis defined by Sepsis-3 criteria.

There are some limitations to this study. The enrolled participants were from a single center, despite random grouping into modeling and validation groups. Thus, external data from another center is needed to confirm the efficiency of our established predictive model. Moreover, bias in treatment and diagnosis could not be avoided in this study because of the large sample size and retrospective nature, although the same inclusion and exclusion criteria were adopted in data selection for all enrolled cases. In addition, the prediction power of our model might be influenced among the patients with septic shock due to differences in management and outcomes.

Conclusion

A simple and applicable model for accurately predicting the risk of requiring mechanical ventilation within 48 h after admission in patients with sepsis was established by combining six clinical indicators. This model can assist in clinical decision-making to manage and treat hospitalized patients with sepsis.

Supplemental Information

Supplemental Information 1 The machine learning models based on C5.0, SVM, XGboost and ensemble methods in the modeling group.

Supplemental Information 2 The importance of variables in the machine learning models.

Supplemental Information 3 The VIFs of variables in modeling group.

Supplemental Information 4 The P values between variables and logitp by boxTidwell test.

Supplemental Information 5 Code.

Crp, high sensitivity -C reactive protein; Alt, Alanine transaminase; Tg, Triglyceride; Tbi, Total bilirubin; Cr Creatinine; Lac, Lactic acid; Bnp, pro-brain natriuretic peptide; Che, Cholinesterase; Pt, Prothrombin time; D.d, D-dimer; K, Potassium; Na, Sodium; Mg, Magnesium; Ca Calcium; Wbc, White blood cell; Hbg, Hemoglobin; Plt, Platelet; Glob, Globulin; Spo2, pules oxygen saturation; T, Temperature; Map, mean arterial pressure; Hp, Heart rate ;R, Breathe rate; GCS, Glasgow coma score; Clugd, Chronic lung disease; Chd, Chronic heart disease; Cld, Chronic liver disease; Ckd, Chronic kidney disease; Head, Intracranial infection; Lung, Lung infection; Bilinay, Biliary infection; Urinary, Urinary infection; Gastro, Gastrointestinal infection; Sofa, Sequential organ failure assessment; News, Nation Early Warning Score.

Supplemental Information 6 Data.

Crp, high sensitivity -C reactive protein; Alt, Alanine transaminase; Tg, Triglyceride; Tbi, Total bilirubin; Cr Creatinine; Lac, Lactic acid; Bnp, pro-brain natriuretic peptide; Che, Cholinesterase; Pt, Prothrombin time; D.d, D-dimer; K, Potassium; Na, Sodium; Mg, Magnesium; Ca Calcium; Wbc, White blood cell; Hbg, Hemoglobin; Plt, Platelet; Glob, Globulin; Spo2, pules oxygen saturation; T, Temperature; Map, mean arterial pressure; Hp, Heart rate ;R, Breathe rate; GCS, Glasgow coma score; Clugd, Chronic lung disease; Chd, Chronic heart disease; Cld, Chronic liver disease; Ckd, Chronic kidney disease; Head, Intracranial infection; Lung, Lung infection; Bilinay, Biliary infection; Urinary, Urinary infection; Gastro, Gastrointestinal infection; Sofa, Sequential organ failure assessment; News, Nation Early Warning Score.

Additional Information and Declarations

Competing Interests

Author Contributions

Human Ethics

Data Availability

The authors declare that they have no competing interests.

Bin Wang conceived and designed the experiments, performed the experiments, analyzed the data, prepared figures and/or tables, authored or reviewed drafts of the article, and approved the final draft.

Jian Ouyang performed the experiments, analyzed the data, authored or reviewed drafts of the article, and approved the final draft.

Rui Xing analyzed the data, authored or reviewed drafts of the article, and approved the final draft.

Jiyuan Jiang analyzed the data, prepared figures and/or tables, and approved the final draft.

Manzhen Ying conceived and designed the experiments, authored or reviewed drafts of the article, and approved the final draft.

The following information was supplied relating to ethical approvals (i.e., approving body and any reference numbers):

The Ethics Committee of Dongyang People’s Hospital.

The following information was supplied regarding data availability:

The raw data is available in the Supplemental File “data.csv” and code file is available in the Supplemental File “Rhistory.R”.

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
