# Peer review of "A novel nomogram to predict the risk of requiring mechanical ventilation in patients with sepsis within 48 hours of admission: a retrospective analysis"

_PeerJ, doi:10.7717/peerj.18500_

## Round 0.1 · original submission · Major Revisions

1. Please Improve clarity and readability:
a. Provide more detailed background information, especially when introducing b. SOFA/NEWS scores and machine learning models
c. Be consistent with notation and correct minor typos (e.g. SPO2, D-dimer)
d. Split longer sentences to improve flow
e. Mention full names before using acronyms
2. Please strengthen the experimental design:
a. Provide more details on the development of the machine learning models, including the techniques used
b. Clarify if the machine learning models were trained on the modeling group data before being tested on the validation group. If not, the models should be trained first on the modeling group to establish credibility. Report findings for both groups.
c. Explain the implications of Figure 6B in assessing model calibration
3. Please justify the novelty and utility of the nomogram:
a. Most of the predictors in your model (lactic acid, pro-BNP, albumin, prothrombin time, lung infection, D-dimer) are well-known to clinicians. Explain what makes your nomogram "novel" compared to existing knowledge and tools.
b. Consider reformatting the nomogram (e.g. as a formula) to make it easier for busy clinicians to use. The current figure may not be immediately usable.
c. Discuss how you balanced model simplicity vs accuracy, and address potential overfitting concerns
4. Use standard expressions and formatting consistently (e.g. ALT as xULN)
and justify inclusion/exclusion of certain variables that may not be routinely assessed or closely related to mechanical ventilation
5. Table 1: Use mean±SD for normal continuous variables, ALT as ULN, consistent p-value decimal places
6. Clarify inconsistency between line 134 and Figure 1
7. Provide missing supplementary tables S1 and S2
8. Double check figure callouts (Fig 6A/B seem swapped)
9. Explain the OR of 1.000 for pro-BNP and if it should be an independent risk factor given this

Reviewer 1 ·

Basic reporting

no comment

Experimental design

no comment

Validity of the findings

no comment

Additional comments

Thank you for inviting me to review this study focusing on the risk prediction of mechanical ventilation. The study raise a novel nomogram and verify its effectiveness and clinical applicability. I thank the authors for their great work, few points need to be clarified.
1. Some expressions and format in the article should be more standardized and unified. e.g. (line 63/line80 SPO2, line 77/line80/Table 1 hs-CRP, MAP)
2. Some diseases or status are not routine and may not be closely related to mechanical ventilation which could be excluded. e.g. chronic liver disease,digestive tract infection, biliary tract infection, urinary tract infection
3. The authors use multiple collinear analysis to analyze risk factors(line 96).Mechanical ventilation or not ,as a categorical variable,should be analyzed by Binary logistic regression analysis.
4. Line 134(After excluding 150 patients requiring mechanical ventilation at admission) is not consistent with Figure 1, please make sure.
5. Regarding Table1,a) normally distributed continual variables should be expressed as mean±standard deviation.b) alanine transaminase should be expressed as ULN. e.g. ALT(>ULN) n(%).c)the decimal places of the p-value should be uniform.Similar issues in Table 2.
6. I don’t find TableS1 and TableS2?
7. How to explain the OR of Pro-BNP is 1.000 with a remarkable p-value?Is it reasonable to consider it as an independent risk factor?
8. Figure6A and Figure6B mark in a wrong place in the text(line 193-196).

Reviewer 2 ·

Basic reporting

The manuscript is written in unambiguous, professional English, conveying the information effectively. It provides sufficient literature references and contextual background, supporting the study's relevance. The article is well-structured, with appropriate figures and tables, and includes raw data that aligns with the hypotheses, making it a self-contained and comprehensive study. The potential impact of these findings on the field is significant. Splitting longer sentences would improve the clarity. Additionally, detailed background information would improve the article's readability and flow. For example:
Line 46: briefly explain the SOFA and NEWS scores and why and how they have been used. You can use the complete form and then start using acronyms.
Line 63: Consistency in the notation, such as SPO2.
Line 78, 146: "D-dime" should be corrected to "D-dimer"
Line 119: Mention the full names of the machine learning models, such as decision tree, Extreme gradient boosting, and Support Vector Machine. Then, use the acronyms.
Line 151: Mention the name of both logistic and stepwise regression
Line 226: typo r.

Experimental design

The manuscript presents original primary research that falls well within the Aims and Scope of the journal, addressing a critical need in predicting the short-term requirement for mechanical ventilation in sepsis patients. The research question is clearly defined, relevant, and meaningful, as it aims to fill a significant knowledge gap by developing a predictive model where none previously existed. The investigation is conducted rigorously, adhering to high technical and ethical standards. The authors provided detailed information on methods to allow other researchers to replicate the study accurately. However, there is scope for improvement by providing details on developing the machine learning models. The authors touched upon the topic without adequately explaining the method or techniques. For example:
Line 119/120: More background information on each machine-learning model is required.
Line 189: Were the machine learning models, the decision tree, gradient boosting, SVM, and ensemble methods used only on the validation group? Were these models constructed/trained on the modeling group data? If yes, why is there no report on these machine-learning models in the modeling group? It’s essential to establish the credibility of the machine learning models by reporting their training on the modeling group, and testing these models to the validation group is essential. If the modeling group data was not used to train these models, then the authors need to train them on the modeling group first and then test it on the validation group. Without the training information, the findings from these machine learning methods lack credibility.
Line 196-198: A detailed explanation is needed for line 196 (Figure 6B) to explain better the implications of displaying the graph between observed and predicted event percentages (Figure 6B) to assess the calibration of the predictive model.
Figure 6: Provide figures for both modeling and validation groups.

Validity of the findings

The manuscript presents statistically sound, robust, and well-controlled findings, with all underlying data provided to support the conclusions. The conclusions are clearly stated, directly linked to the original research question, and appropriately limited to the results obtained, avoiding overreaching claims. The study does not rely on subjective assessments of impact or novelty, allowing for meaningful replication and further validation by the research community. Replication studies are encouraged, provided they clearly state their rationale and contribution to the literature. The research adheres to high data transparency and integrity standards, making it a strong contribution to the field. However, not reporting the training of the models questions the credibility of the models and, thus, the study findings.

Additional comments

The manuscript presents a well-conducted and significant study that addresses a critical gap in the clinical management of sepsis patients. The development and use of a predictive model in clinical practice for the need for mechanical ventilation within 48 hours of admission is both timely and relevant, given the limitations of existing tools like the SOFA and NEWS scores. However, the manuscript will become scientifically robust if additional changes are made to train the machine learning models using modeling data and then validate the models on the validation group. Reporting training and validation of models is vital if the authors are willing to use machine learning techniques to compare with the regular models. In that case, the manuscript will become well-structured, with a logical flow and a clear and valid presentation of results.

·

Basic reporting

Acceptable.

Experimental design

Acceptable.

Validity of the findings

Acceptable.

Additional comments

1. To predict 48-hour mechanical ventilator in sepsis patients, your model has the predictors of: lactic acid, pro-BNP, and albumin levels, prothrombin time, presence of lung infection, and D-dimer level. Most of them are well known to practicing clinicians. Please explain why your nomogram or model is "novel". Your predictors need blood lab and images and so are not so convenient or immediate.
2. Your nomogram is on Figure 2. Please make it easy to use for busy clinicians. I, myself a clinician, am unable to use your nomogram on the first sight. How about you make it like a formula: Y= B0 + B1*X1 + B2*X2 +...+ Bn*Xn. This form may be easier to use?
3. How do you trade off simplicity and AUC for your prediction model. How could you make your nomogram simple and easy but also accurate enough? How about over-fitting? Please explain.

---

## Round 0.2 · Minor Revisions

1. Discuss how variations in sepsis definitions among other studies were handled to address concerns about heterogeneity.
2. Clarify the standardization of different mortality definitions and address competing causes of death.
3. Provide a clearer statement on how to simplify the nomogram model for practical use by busy clinicians, focusing on the practicality of using a single indicator versus multiple variables.
4. Strengthen the justifications on how the study uniquely adds to existing knowledge beyond the known factors.
5. Provide a detailed explanation of how the cutoff points for variables like lactate and SOFA were validated.
6. Please consider including an intensivist's input to enhance the manuscript's clinical rigor, particularly in ICU settings.
7. Please improve the manuscript's readability and flow through additional language editing.
8. Please address septic shock as a distinct entity in future revisions, given the potential differences in management and outcomes.

Reviewer 1 ·

Basic reporting

no comment

Experimental design

no comment

Validity of the findings

no comment

Additional comments

Thank you for your effort. I have no other comments

·

Basic reporting

Sepsis Definition Consistency: The document defines the sepsis criteria used in the study as Sepsis 3.0 (page 6), which is a well-established, standardized definition. However, it does not discuss how variations in sepsis definitions among other studies were handled, which may still leave concerns about heterogeneity unaddressed. A stronger explanation of how these differences were managed would improve consistency across studies.

Mortality Definition: The document primarily focuses on mechanical ventilation rather than various definitions of mortality (e.g., 30-day mortality, hospital mortality). The absence of a clear discussion on standardizing different mortality definitions and addressing competing causes of death (e.g., cardiac death unrelated to sepsis) still remains. This is a point that should be further clarified.

Experimental design

Simplification for Clinicians: The manuscript introduces a nomogram model based on multiple variables (e.g., lactate, pro-BNP, D-dimer). While these are clinically relevant indicators, the study could benefit from a clearer statement on how to simplify this information for practical use by busy clinicians. The authors should focus more on the practicality of using a single indicator versus multiple variables in clinical settings.

Study Rationale: The motivation for the study—developing a model to predict mechanical ventilation in sepsis patients—addresses an unmet need by targeting respiratory failure within 48 hours. However, the manuscript could still provide stronger justifications on how it uniquely adds to existing knowledge beyond the known factors (e.g., lactate, SOFA).

Validity of the findings

Cutoff Point Explanation: The cutoff values for variables like lactate and SOFA are based on clinical judgment and past studies, but it is unclear if ROC analysis was specifically performed to identify the 2.0 cutoff (page 7). A detailed explanation of how these cutoff points were validated would address this concern.

Heterogeneity in Meta-Analysis: The manuscript does not seem to include a meta-analysis. It’s a single-center retrospective cohort study. Thus, the issue of heterogeneity across studies does not apply directly to this paper. However, the authors could improve the external validity by discussing potential variability in sepsis definitions and clinical practices in different centers.

Additional comments

Intensivist Involvement: There is no mention of an intensivist in the author list or acknowledgments. Including an intensivist's input would enhance the manuscript’s clinical rigor, particularly in ICU settings.

English Editing: The manuscript is generally clear but could benefit from additional language editing to improve readability and flow.

Septic Shock Discussion: The manuscript focuses primarily on sepsis and mechanical ventilation, without a separate discussion on septic shock. Given the potential differences in management and outcomes, it may be useful to address septic shock as a distinct entity in future revisions.

In conclusion, the revision addresses some of my concerns, particularly regarding the use of Sepsis 3.0 as a definition and the nomogram model for predicting mechanical ventilation. However, issues regarding the standardization of mortality definitions, further simplifying the model for practical use, and including an intensivist’s review remain to be addressed.

---

## Round 0.3 · accepted · Accept

All comments have been fully addressed by authors. They have added the content to the introduction, changed the nomogram style, described more about machine learning techniques. They also reedited the language and revised the image, and thus I think this paper has been significantly improved. This paper can be accepted for publication.